# Membrane Water Treatment for Drinking Water Production from an Industrial Effluent Used in the Manufacturing of Food Additives

**DOI:** 10.3390/membranes12080742

**Published:** 2022-07-29

**Authors:** Karina Hernández, Claudia Muro, Oscar Monroy, Vianney Diaz-Blancas, Yolanda Alvarado, María del Carmen Diaz

**Affiliations:** 1Tecnológico Nacional de México/Instituto Tecnológico de Toluca, Avenida Tecnológico S/N Colonia Agrícola Bellavista, Metepec, Ciudad de México 52140, Mexico; khernandezg@toluca.tecnm.mx (K.H.); vdiazb@toluca.tecnm.mx (V.D.-B.); yalvaradop@toluca.tecnm.mx (Y.A.); mdiazn@toluca.tecnm.mx (M.d.C.D.); 2Departamento de Biotecnología, Universidad Autónoma Metropolitana, Avenida San Rafael Atlixco 186, Colonia Vicentina, Iztapalapa, Ciudad de México 09340, Mexico; monroy@xanum.uam.mx

**Keywords:** food effluents, water recovery, potable water, membrane treatment, saline effluents, dye removal

## Abstract

An integrated membrane process for treatment of effluents from food additive manufacturing was designed and evaluated on a laboratory scale. The principal focus was water recovery with the possibility of its reuse as potable water. The industrial effluent presented high content of dyes and salts. It was red in color and presented brine characteristics. The whole effluent was fed into the integrated process in continuous flow. The steps of the process are as follows: sedimentation (S), adsorption by activated carbon (AC), ion exchange using resins (IEXR), and reverse osmosis (RO) (S–AC–IEXR–RO). The effect of previous operations was evaluated by stress-rupture curves in packaged columns of AC and IEXR, membrane flux, and fouling dominance in RO. Fouling was evaluated by way of the Silt Density Index and membrane resistance examination during effluent treatment. The integrated membrane process provided reclaimed water with sufficiently high standards of quality for reuse as potable water. AC showed a high efficiency for color elimination, reaching its rupture point at 20 h and after 5L of effluent treatment. IEXR showed capacity for salt removal, providing 2.2–2.5 L of effluent treatment, reaching its rupture point at 11–15 h. As a result of these previous operations and operating conditions, the fouling of the RO membrane was alleviated, displaying high flux of water: 20–18 L/h/m^2^ and maintaining reversible fouling dominance at a feed flow rate of 0.5–0.7 L/h. The characteristics of the reclaimed water showed drinking water standards

## 1. Introduction

The scarcity of potable water and its recovery by water treatment is a great environmental concern which currently occupies agendas in many countries [1]. In particular, residual and contaminated effluents from the industrial sector urgently need effective water recovery systems and reuse because water scarcity and water pollution could limit growth and production in coming years, affecting the world economy and human heath [2].

Water recovery by effluent treatment is a sustainable process, providing great environmental benefits, and significant monetary savings. Owing to this fact, water is fast becoming a resource to be reclaimed. By reclaiming water, aquifer overexploitation is reduced, more people will have access to potable water, and the financial burden of disposing of or treating polluted effluents will be minimized. This fact makes water recovery an attractive option for the industry and the environment.

Currently, there are different methods of water decontamination; however, the application of sequential operations to ensure clean water production and reuse is essential for water recovery [3]. In this field one finds hybrid/integrated membrane processes, which consist of previous operations of depuration and membrane technologies, such as microfiltration (MF), ultrafiltration (UF), nanofiltration (NF), and reverse and forward osmosis (RO and FO) [3,4,5].

As a result of the complexity of industrial effluents and their depuration necessities, hybrid membrane systems are designed and applied as integral processes to make effluent cleaning and water recovery more efficient and sustainable processes.

In addition to membrane technologies, other conventional processes may be applied, including sedimentation, coagulation, precipitation, filtration, biological and chemical oxidation, adsorption, and ion exchange. They are strategically integrated and organized as previous processes or pretreatments to increase membrane productivity and provide better results of depuration and water recovery, mitigating membrane fouling. Moreover, there are other sequential membrane procedures, such as NF/RO and MF/RO, which are integrated for a similar purpose. According to effluent characteristics, synchronized membrane operations, such as membrane distillation (MD) and membrane bioreactors (MBR), may be integrated in these processes.

At present, several combinations of membrane technologies are possible, as well as the integration of hybrid systems of depuration. However, their application requires numerous studies because it depends on several factors, such as effluent characteristics, the required quality of the reclaimed water, membrane efficiency, and the cost of the treatment process to achieve the necessities of effluent treatment and water recovery.

The principal function of membranes is the exclusion and isolation of different water pollutants. They permeate clean water with the required standards for reuse. Principally, RO, FO and MD are applied to desalination effluents for the purposes of ion removal and separation of dissolved compounds <100 Da (<1 nm). This produces reclaimed water of a high standard, providing the possibility of recycling it as potable water, boiler water, and/or cooling water [6,7]. In turn, NF membranes are used for separating effluent constituents <1000 Da (<10 nm), such as active substances, metals, surfactants, colorants, foods, oil and gas, pharmaceutical compounds, as well as pulp and paper. Doing so also provides a high quality of reclaimed water [8,9,10]. Other processes, such as UF and MF membranes, are used to remove suspended solids, colloidal matter, bacteria, and viruses. Water recovered in this way has great potential for reuse [11].

Recent publications provide positive data on the application of integrated processes for effluent treatment. This information is available in Zhang et al. [12]. They compared two treatment processes for the depuration of a biologically treated effluent from a wastewater treatment plant (WWTP). The authors tested UF and electrochemical–microfiltration–granular activated carbon adsorption (e–MF–GAC) as pretreatment of RO for fouling control. The best results were provided by e–MF–GAC.

Ćurić and Dolar [13] tested sand filtration (SF), coagulation, and coagulation/flocculation (C, C/F) as individual pretreatments before UF for the depuration of effluents from washing dyeing machines. In this case, SF proved to be more efficient, producing water for reuse.

Partal et al. [14] developed an integrated membrane process for water and salt recovery. The effluent in their study was from a textile mill. They used UF–RO to obtain reusable water, whereas the brine stream was treated by ozone oxidation, NF, RO and IEX (OO–NF–RO–IEX), with zero liquid discharge. Chen et al. [15] developed an integrated membrane process by adsorption, coagulation, and RO (AD–C–RO) for the treatment of textile effluents and wastewater reclamation. The alleviation of RO membrane fouling led to successful treatment. Günes and Gönder [16] used a membrane system which included electrocoagulation–NF–RO (EC–NF–RO) for the treatment of a textile effluent and achieved high water production and high quality for its use in textile finishing processes.

Hernández et al. [5] treated a food effluent by UF–RO. The procedure resulted in reversible fouling and clean water production for reuse. Chandrasekhar et al. [17] used an integrated membrane system with a bioreactor (IMBR)–UF–RO for the treatment of an industrial coffee effluent. The recovered water was of a sufficiently high quality for industrial reuse. Lebron et al. [18] evaluated the treatment of vinasse effluents by coagulation–MF–NF (C–MF–NF); the NF permeate quality indicated reuse possibilities, such as washing water and cooling water. Dos Santos et al. [19] treated cassava starch wastewater by coagulation/flocculation–MF (C/F–MF), improving the final quality of the treated water. Czuba et al. [20] used an integrated process involving MF–UF–NF–RO to recover nutrients and water for use in several applications.

The results found in previous investigations show the versatility of integrated membrane processes for the treatment of different type of effluents, including several combinations of operations prior to RO to reduce fouling and increase membrane efficiency. However, to meet the objective of treating and recovering water, the design and study of the integrated membrane process are necessary to determine the adequate operations and process conditions that improve the membrane use and data for process scaling.

In this study, an integrated membrane process involving RO was designed and evaluated to create an efficient effluent treatment solution for water recovery.

The effluent comes from the manufacture of food additives with high concentrations of dyes and salts. The suggested operations prior to the RO process were sedimentation (S), adsorption by activated carbon (AC), and ion exchange, using resins (IEXR). Dynamic studies of AC and IEXR were included to provide data on operating conditions and the effect on RO fouling. The standards of reclaimed water were established in accordance with industrial needs and potable water values. The obtained information could be used to scale the process to pilot or industrial scale.

## 2. Materials and Methods

### 2.1. Materials

Reagents H_2_SO4 (Jalmek), Hg_2_SO_4_ (J.T.Baker), Ag_2_SO_4_ (J.T.Baker), K_2_Cr_2_O_7_ (J.T.Baker), and K_2_CrO_4_ (J.T.Baker) were used for determining COD. AgNO_3_ (J.T.Baker) was used to determine Cl^−^ concentration. OH (Jalmek) and HCl (J.T. Baker) were used for IEXR regeneration.

### 2.2. Physicochemical Characterization of the Industrial Effluent

A total of 20 L of industrial effluent was collected at the discharge point of an industrial plant located in Mexico. The effluent is wastewater from a manufacturing process used for creating food additives. The collected sample was stored in 1 L containers and kept in refrigeration at 4 °C for future analysis.

In accordance with the procedures described in Standard Test Methods for Wastewater [21], the effluent was characterized in terms of chemical parameters, including Chemical Oxygen Demand (COD), Electrical Conductivity (EC), Chlorides (Cl^−^), Total Solids (TS), Settleable Solids (SS), Total Dissolved Solids (TDS), and Coloration (absorbance to maximum band). The equipment used was a Perkin Elmer UV/VIS Spectrophotometer, Lambda 35 (Waltham, MA, USA), a HANNA benchtop multiparameter meter (brand HI5522-01), and a potentiometer with a pH HI 1131B electrode BNC, using five points of calibration.

Table 1 shows data regarding the chemical characteristics of the industrial effluent and the standard deviation of these values as a result of averaging three samples and three determinations.

The effluent is an intense red color, it has basic pH values and high values of EC and TS. These parameters were associated with preeminent TDS concentration, revealing excessive salt and colorant content. COD was principally associated with the presence of residual organic colorants.

Figure 1 shows a spectrophotometric Uv-vis scan (200–800 nm) of the industrial effluent. 

Uv-vis spectra confirmed the predominance of red coloration in the effluent, displaying a maximum band at 490 nm with high absorbance (A = 3.5 at 10% of effluent dilution). The strong coloration was associated with the content of red colorants in the mixture, such as red No. 2 (540 nm), red No. 3 (530 nm), and red No. 40 (520 nm). In addition, a small band at 430 nm was observed, indicating the presence of yellow dyes, such as yellow No. 5 (430 nm), which explains the shift to the left of the band of red dyes. Spectra also showed bands in the UV range at 320 nm, indicating the presence of azo groups. In turn, bands at 200–300 nm may indicate aromatic groups and salt content in the effluent.The presence of these colorants in the effluent was confirmed by scanning synthetic water containing a single component, a mixture of these colorants, and salt. Maximum absorbances for single components were as follows: red 40 (520 nm), red 2 (504 nm), and yellow 5 (390 nm). The maximum wavelength of the mixture of synthetic dyes and salt indicated a maximum band at 500 nm. The differences in spectra of the industrial effluent and the synthetic wastewater were attributed to the presence of salt, which caused shifts to the right and left, according to the spectrum of each colorant. This fact was associated with the aggregation of the dye or dyes with the salts present in the effluent.

Thus, industrial effluent composition and coloration were linked to wastewater from the industrial synthesis of the following food azo dyes: red (No. 2, 3 and 40) and yellow (No. 5).

Food azo dyes are obtained by reacting aromatic amines (Ph-NH_2_) with nitrous acid HNO_2_ (O=N-OH) (prepared in situ using NaNO_2_ and HCl) to obtain a nitrogenated compound called aromatic diazonium salt (Ph-N=N:). As a byproduct of this chemical reaction, common salt (NaCl) is obtained. In turn, aromatic diazonium salts react with phenols or arylamines by electrophilic aromatic substitution (coupling), producing (red or yellow) azo dyes in basic conditions of pH.

Figure 2 shows the chemical reactions involved in the production of food azo dyes, and the chemical structure of dyes.

Salts are produced by the synthesis of food azo dyes. NaCl as Na^+^ and Cl^−^ ions, stable salts, and other inorganic and organic salts predominate. Therefore, the synthesis of azo dyes accounts for the high content of TS, TDS and the high EC in the effluent. Furthermore, the basic pH of the effluent is a result of the synthesis of azo dyes.

Based on the aforementioned observations, the industrial effluent was identified as colored wastewater with strong red coloration. The high TDS content also characterizes the effluent as ‘high-TDS brine’ [22]. NaCl prevailed. The components found in the industrial effluent were associated with the remaining red and yellow dye mixture from food dye production. Pollution and uncommon biodegradation were caused by the high salt content and the presence of azo dyes.

### 2.3. Membrane Treatment Process Applied to Industrial Effluent for Water Recovery

Due to the presence of dyes and effluent salinity, an integrated RO membrane process was designed to reduce the effluent’s principal physicochemical characteristics and produce clean water for reuse. The integrated process consisted of four separation units. (1) Sedimentation for settleable solids removal (S), (2) adsorption for coloration removal, using activated carbon (AC), (3) ion exchange for ionized salts removal (Na^+^ and Cl^−^), using resins (IEXR), and (4) reverse osmosis (RO) as the final step for water recovery and salt (NaCl) concentration.

The proposed treatment process was evaluated at laboratory scale, under the following conditions: sedimentation was tested for 2–4 h. The adsorption process was carried out in a glass adsorption column (0.15 m^2^) using granular AC F300 mesh (effective size 0.8–1.0 mm) as adsorbent material. Ion exchange by resins (IEXR) was carried out in glass columns (0.15 m^2^). Resins of polystyrene were used as packaging material for ion exchange (cationic H^+^, 1.7 meq/mL and anionic OH^−^, 0.9 meq/mL).

Next, a roll of polyamide membrane with an area of 0.53 m^2^ was used for RO in continuous mode at 9 bar of pressure.

The AC and IEXR were packed into their respective columns. Prior to effluent treatment, the packaging materials were washed with distilled water to remove impurities; subsequently, resins were humected with deionized water for 48 h.

The treatment process was carried out in continuous flow at the pH of the effluent. According to the bed volume of the columns (240 mL), a feed volume of 200 mL of the industrial effluent was indicated as one experimental run of the effluent treatment.

Figure 3 describes the process of effluent treatment. A container (1) was used to store the effluent. A peristaltic pump (2) and pipeline (3) were used to transport and feed the settler (4). Two streams were identified at the outlets of this separation unit, a supernatant (4F), and a product of settleable solids (4S). Stream 4F was fed into the adsorption column (5). Subsequently, the outlet effluent from the adsorption column (5F) was fed into the ion exchange columns; first into the cationic resin (6) and then into the anionic resin (7). The outlet streams of these columns are represented as 6F and 7F. In turn, stream 7F was fed into the osmosis unit (8), where two manometers were placed at the entrance and exit points to measure inlet pressure P1 (9) and outlet pressure P2 (10). A valve (11) was used to control pressure; pipelines (12) and (13) were used to transport the rejected salt (8R) and permeate (water recovery) (8P) from the osmosis membrane to their storage containers, (14) and (15) respectively.

The number of separation units, their dimensions, and the operating conditions during the treatment process are detailed in Figure 3. Due to the high salinity and presence of Cl^−^ ions in the industrial effluent, the ion exchange operation made use of three columns, one for cation removal, Na^+^, and two for anion removal, Cl^−^. The columns were identified as I and II. Flows in sedimentation, adsorption and ion exchange columns were previously established.

In each run of the treatment process, samples from separation units were obtained to determine the quality parameters of the treated effluent. The quality parameters and their determination were established according to the physicochemical characteristics of the industrial effluent and the procedures described in Section 2.1. COD, TDS, and EC data were used as principal parameters for monitoring the removal percentage of coloration and salts from the industrial effluent.

### 2.4. Experimental Determination of Rupture Curves in Adsorption and Ion Exchange Columns

Effluent feed volumes and operating times of treatment were registered until AC and IEXR reached saturation. Data were used to obtain experimental rupture curves, providing the behavior of packaging materials during the treatment by adsorption and IEXR in each column, respectively.

Volumes of the treated effluent at rupture points were registered as V_b_, while operating times were identified as θ_b_ (breakthrough time) of adsorption. Ion exchange columns were used to calculate the maximum capacity of materials for component removal at rupture point (q_b1_). The mass of component removal per volume of packaging material (mg/mL) was calculated by Equation (1). The maximum capacity of materials for component removal at rupture point (q_b2_) and the mass of component removal per packaging material mass (mg/g) were calculated by Equation (2):(1)$$q_{b1}=\frac{V_{b}\left(C_{o}-C_{b}\right)}{\mathrm{Vm}}$$
(2)$$q_{b2}=\frac{Q{Ɵ}_{b}\;C_{o}}{m}$$

Effluent concentrations were associated with COD (mg/L) to describe color removal in the adsorption column. The TDS (mg/L) parameter was used to define salt removal in ion exchange columns. C_0_ was assigned as the initial concentration of the effluent (mg/L) and C_b_ as concentration at the θ_b_ of the treated effluent (mg/L). Q corresponds to volumetric flow in the columns. The bed volume or bed mass of the packaging material in the columns was assigned as V_m_ = 240 mL, equivalent to mass material = 127 g of AC, cationic IEXR = 288 g, and anionic IEXR = 257 g.

In turn, saturated materials were regenerated as follows. AC was subjected to pyrolysis at 950 °C for 2 h with a heating rate of 10 °C/min. Then, distilled water was used for carbon washing, removing the ashes. The material was dried and stored for later use.

Anionic and cationic IEXR were washed with deionized water at room temperature; regeneration was carried out in NaOH solution (1% *w*/*v*) and HCl solution (1% *v*/*v*), respectively.

Regenerants and washed water were fed via countercurrent flow (carbon 4 mL/min and resins 2 mL/min). Color, EC and Cl^−^ were used as indicators of carbon and resin regeneration, respectively.

Likewise, the status of regenerated resins was monitored by morphological analysis and the elemental composition of surface resins. Scanning Electron Microscopy (SEM) and Energy Dispersive X-ray Spectroscopy were applied using a JEOL, JSM-6610LV Microscope (manufacturated by JEOL Ltd., Tokyo Japan). Resin samples were covered with a thin layer of gold. The wattages used for analysis were 5, 10, and 15 kV.

### 2.5. Determination of Effectiveness of RO Membrane for Water Recovery from the Industrial Effluent

Membrane efficiency for water recovery was evaluated by way of feed flow (0.5, 0.7, and 1.0 L/min), analyzing water permeate fluxes, salt rejection percentages, and fouling dominance, using the Silt Density Index (SDI) and membrane resistance examination on flux decline during effluent treatment [5].

The membrane was tested in crossflow mode, using continuous flow with transmembrane pressure (TMP) of 9 ± 0.1 bar. The permeate and reject currents were separated from the membrane (without recirculation), as is observed in Figure 2.

Water permeate fluxes (J, L/h/m^2^) were determined by the volumetric flow of permeate water per membrane area (0.53 m^2^). The removal efficiency of the membrane was determined by rejection percentage (R_s_%), using Equation (3). C_p_ was assigned as TDS concentration in permeate water and C_o_ as concentration of TDS in the effluent fed to the membrane (the effluent treated in the II resin anionic column):(3)$$R_{s}\%=\frac{\left(C_{o}-C_{p}\right)}{C_{o}}\;\times 100$$

SDI was determined by the ASTM D 4189-95 standard test method to ascertain the effect of the salt’s quantity in the feed effluent on the membrane’s capacity for salt rejection and water permeation [5].

Fouling dominance in the membrane was evaluated by membrane resistances to water flux, including (1) resistance following wastewater treatment (R_t_) with water flux as J_p_. (2) Hydraulic resistances, using initial membrane resistance of clean water flux (J_w_) prior to wastewater treatment (R_m_), and membrane resistance of water flux (Jwi) after the removal of fouling through washing the membrane (R_ir_) (irreversible deposition onto the membrane surface or adsorption on the membrane surface). (3) Resistance due to reversible fouling (concentration polarization and/or reversible deposit) (R_r_). Equations (4)–(7) were used to calculate R_t_, R_m_, R_ir_, and R_r_. The viscosity of the effluent and clean water at 20 °C were assigned as µ_e_ = 1.564 cp and µ_w_ = 1.0019 cp respectively [5].

All experiments and analytical determinations were conducted in triplicate. Data were processed by statistical methods in the Origin software, version 2016 (9.3), created by OriginLab Corporation (Northampton, MA, USA):(4)$$t=\frac{\mathrm{TMP}}{{µ}_{e}J_{p}}$$
(5)$$R_{m}=\frac{\mathrm{TMP}}{{µ}_{w}J_{w}}$$
(6)$$R_{\mathrm{ir}}=\frac{\mathrm{TMP}}{{µ}_{wi}J_{\mathrm{wi}}}$$


R_r_ = R_t_ − R_ir_ − R_m_
(7)


## 3. Results

### 3.1. Water Recovery by Industrial Effluent Treatment

Table 2 provides data on the physicochemical characteristics of the effluent treated by the integrated process described in Section 2.3. Data correspond to first treated volume (recovered water) = run 1 = 200 mL, operating in the separation conditions indicated in Figure 2. The standard deviation of the data is relative to the average value of three samples and three determinations for each experiment.

All separation units achieved a high efficiency, allowing fresh water production following treatment. The treated effluent showed a particularly drastic reduction in the SS parameter following sedimentation. This indicates the total removal of settleable solids (a concentration of settleable solids was not detected in the Imhoff cone).

In turn, the effluent from the adsorption column registered null color absorbance at 490 nm in Uv-vis spectra, exhibiting entire coloration removal. Elimination of the coloration also caused a COD reduction of 99.5%, demonstrating its relationship with the presence of dyes in the industrial effluent. As expected, other parameters, such as EC, TS, and TSS, were not reduced by adsorption using AC.

Then, the treated effluent from AC (saline effluent) was passed through cationic IEXR, where the H^+^ was replaced by Na^+^. Then, the cationic treatment effluent was passed over anionic IEXR (columns I and II), replacing OH^−^ with Cl^−^. The release of H+ and OH^−^ ions from resins and their replacement with Na^+^ and Cl^−^ ions from the effluent caused a 70–85% reduction in EC, TSS, TS, and Cl^−^ during the first volume of the treated effluent (200 mL).

To end the treatment process, the effluent treated by resins with 20–30% salt content was fed into the RO membrane, allowing a high percentage of salt removal (99%). As a result, decontaminated water was permeated by the membrane, achieving a sufficiently high quality for its reuse. Furthermore, a salt concentration with the possibility for recycling was produced by this unit. In accordance with the standard regulations for potable water, such as from the World Health Organization (OHO) [23], Environmental Protection Agency (EPA) [24] and Official Mexican Standard NOM-127-SSA1-1994 [25], the chemical quality of water produced by RO was comparable to potable water, showing values below those indicated in the regulations.

Due to the origin of effluent water, metals, pesticides, and other substances were not included in these results because they were not detected before or after treatment. Furthermore, microbiological parameters were not measured in this study because they were not required by the industry; however, RO could theoretically achieve 85% microbial removal. Consequently, the chemical quality standards for industry and potable water indicated that water reclaimed by way of the proposed treatment process can be reused for industrial activities, including boiler and cooling water.

The design of the integrated S–AC–IEXR process as a prior operation to effluent treatment was deliberately developed and applied to reduce dyes and salts because effluents containing both contaminants (especially when dissolved) are difficult to treat. Principally, salts interfere in dye separation, causing reduced efficiencies of color removal, which cause excessive fouling in RO. Nevertheless, adsorption by AC resulted in an effective operation for the separation of azo dyes. AC acted as excellent adsorbent material, with minimum interference because this material is effective in removing dissolved dyes. In this case, sedimentation also contributed to the efficiency of AC and IEXR.

In turn, IEXR was selected as a previous operation to RO for fouling mitigation because the concentration of inorganic salt as stable salts and Na^+^ and Cl^−^ ions in the industrial effluent is excessive; therefore, RO requires a previous operation to reduce these species and the fouling caused by these components. IEXR is often used in conjunction with RO because they remove salts efficiently and are easy regenerable. Owing to this, IEXR is used as pretreatment in several commercial RO systems.

Currently, there exist other conventional methods for reducing salt concentration, such as precipitation and evaporation. However, the presence of stable salts in the effluent reduces precipitation yield. In addition, the accumulation of precipitation agents in the effluent can also further increase membrane fouling; while evaporation increases processing costs, because it requires high energy, while, the water does not recover and its recovery, similarly, requires energy.

Other operations, such as flocculation, coagulation, and adsorption with different adsorbents also were declared as unsuitable processes prior to RO for the separation of organic content in the industrial effluent [21] because they do not work efficiently in the presence of salts. Consequently, an ineffective process for the removal of organic compounds would cause fouling in RO and a reduced efficiency of the treatment. The most well-known contaminants for causing fouling are salts and organic compounds, leading to considerable declines in flux and irreversible fouling. Moreover, prior UF and NF operations could remove salts and dyes; however, both contaminants cause fouling in UF, reducing the efficiency of membranes in the process of treatment.

Based on recent data, it is suggested that an integrated membrane process for water recovery (S–AS–IEXR–RO) by industrial effluent treatment be applied to effluents with comparable characteristics (containing azo food dyes and salts).

Unit operations integrating the process for effluent treatment could also be scaled and combined with different processes to obtain a proposal on a larger scale with a similar purpose.

Usually, food effluents are discharged in wastewater treatment plants (WWTPs) for depuration with domestic wastewaters, using primary and secondary treatment to reduce several parameter indicators of the presence of organic compounds. However, high salinity (in excess of 3000 mg/L) hinders the biological treatment process in a WWTP [26]. In addition, salinity dominates in the treated effluent, and final disposal is regulated for its discharge. Moreover, the quality of depurated water is poor, and it cannot be reused because WWTPs are commonly designed to treat domestic effluents, and to move the treated wastewater to sewers for its discharge.

At present, there are several WWTPs which make use of tertiary processes to obtain recycled water; nevertheless, its reuse is commonly for irrigation because treated water contains organic compounds, azo dyes, and salts. These components are resistant to microbial degradation, stable in light, and the oxidizing agents used in WWTPs [27], limiting water reuse in the industry. However, as a result of the recent emphasis on water recovery, WWTPs are employing new methods to achieve high quality standards and promote water reuse [28].

In turn, degradation and removal of coloration from wastewater are found in numerous contributions to the literature, showing promising results by using different treatment methods, such as microbial cells, adsorption, oxidation, and electrolysis. Recent reviews demonstrate this information [29,30]. However, data are frequently obtained on the treatment of synthetic wastewater with a single azo dye, (other pollutants in the water are not investigated), and a low concentration of dyes is also observed in these reports. Nevertheless, dye separation is affected in industrial effluent treatment because effluents have a complex composition and high pollutant concentrations, causing interferences.

The treatment of saline effluents has also been the focus of numerous research projects. The presence of salts in treated wastewater reduces its feasibility for reuse, including for agriculture and other activities because it causes significant damage to the environment. Here, membrane technology constitutes the most important methods for salt separation [22,31], principally RO and membrane distillation shows strong potential for salt removal and clean water permeation for recycling, including salt recovery [14,32,33,34].

Several reports on industrial effluent treatment and water recovery are described in current reviews [10,28,34]. However, industrial wastewaters containing both dyes and salts have been scarcely studied [31]. Moreover, a high content of these components is not addressed in the literature.

Comparable information on water recovery and membrane use for the treatment of industrial effluents can be seen in [35]. The authors treated textile effluents with a high concentration of Congo red and salts by an MD coupling process, recovering 69% of salts and 48% of clean water. Chen et al. [15] used a sequential process of coagulation–adsorption–RO for reducing fouling in RO, removing dyes and salts from prepared wastewater.

Other data on industrial effluent treatment for water recovery are found in Riera et al. [36]. They used NF to treat effluents from UHT flash cooler condensates from a dairy factory, achieving 87.5% water recovery with economic benefits (TDS were not reported). Buabeng-Baidoo et al. [37] evaluated the profitability of wastewater treatment for the dairy industry, simulating RO process. As a result, they obtained a 33% reduction in freshwater consumption and an 85% reduction in wastewater generation. Gündogdu et al. [38] used NF and RO to treat effluents from MBR (membrane bioreactor; TDS = 1832–1543 mg/L), producing water cooling and boiling feedwaters. Hernández et al. [5], obtained potable water using a sequential process of UF–RO in the treatment of effluents from the protein manufacturing industry (TDS = 2973 mg/L). Cinperi et al. [39] treated a mixture of effluents from the textile industry TDS = 659–5360 mg/L), employing MBR–NF–RO, MBR–NF–UV (disinfection with UV light) and MBR–RO–UV. Reclaimed water was tested in different dyeing recipes at lab-scale. They found that this had no negative effect on quality. Brião et al. [40] proposed an alternative method to treating rinse effluents by RO (COD = 2230 mg/L) from the dairy industry. The produced water was proposed for cooling water use and to be added to fermented milk beverages and caramelized milk.

### 3.2. Rupture Curves in Adsorption and Ion Exchange Columns

Figure 4 describes dynamic studies of AC and IEXR in adsorption and ion exchange columns following 30 and 15 runs of the treatment process for the initial conditions of the materials, and after their regeneration. On the “y” axes, one observes COD to describe color adsorption, and TDS to describe ion exchange from salts in cationic and anionic columns I and II. On the “x” axes, one observes the volume of the treated effluent (L).

According to the operating conditions of adsorption and ion exchange processes, the columns of AC and IEXR presented long rupture times, and high volumes of treated effluent. This indicates good performance in the separation of color and salts from the contaminated effluent. In turn, regenerated materials (AC and IEXR) presented 99% material recovery; therefore, behavior similar to that of initial material conditions was observed in the following runs of effluent treatment.

Table 3 includes information on the capacity for the removal of contaminants from packaging materials during industrial effluent treatment, indicating information on rupture points, such as break concentration (C_b_, mg/L), break volume of the treated effluent (V_b_, L), and break time of columns (θ_b,_ h). The data also include the maximum capacity of packaging materials for color and salts removal at rupture points, which was expressed as qb (mg of colorants adsorbed or exchanged ions/mL of bed material). The concentrations of C_o_ and C_b_ are expressed as COD (initial and rupture point color) for the adsorption column and TDS (initial and rupture points for salts) for resin columns.

AC material confirmed high capacity for color removal during several treatments of the industrial effluent, showing highly efficient COD reduction with a large rupture point. AC saturation was achieved in 25 runs. The rupture point was reached after 20 h of operation of the adsorption column. According to the volume feed (200 mL) into the adsorption column, a bed of 240 mL (bed height of 20 cm) of activated carbon could treat a volume <5 L of industrial effluent, showing an excellent result.

Data on IEXR saturation also showed the capacity for salt removal; however, the salinity of the industrial effluent limited sequential ions separation, achieving smaller volumes and shorter times of separation than AC. Furthermore, the presence of stable salts (undissociated) probably reduced resin capacity for ion (Na^+^ and Cl^−^) removal.

The rupture point in cationic and anionic I columns was found after 10 runs of the treatment process, indicating that a bed of 240 mL (bed height of 20 cm) of resins could treat a volume of <2 L of industrial effluent for 11 h (rupture point), whereas anionic II column reached rupture point at 20 h with a volume of 2.5 L. A reduced flow in this column allowed for a high performance.

Resin regeneration also showed 90–95% efficiency in effluent treatment, indicating high polymer revival. This result was associated with the elimination of settleable and suspended solids by the previous sedimentation operation. Figure 5 shows micrographs of initial, saturated, and partially regenerated IEXR.

Micrographs confirmed (Figure 5b,e) salt accumulation on cationic and anionic resin surfaces during effluent treatment. In addition, elemental composition analysis of saturated resins showed abundant percentages of Na and Cl elements in comparison with virgin IEXR, indicating the principal foulants of IEXR.

The accumulation of salts on resin surfaces reduced the size of the ion exchange zone; causing rupture times during effluent treatment. However, salt removal by resins was recognized as excellent previous operation to RO because the salt concentration of the effluent is high.

Micrographs also showed that IEXR are recovered after their regeneration (Figure 5c,f), exhibiting a clean surface (IEXR are materials of easy and fast regeneration). Therefore, IEXR were established as satisfactory materials for ion removal from industrial effluents, providing high efficiencies of separation. Moreover, operating conditions in IEXR and number of columns of anionic IEXR also contributed to this result.

Consistent with the information regarding dynamic studies in AC and IEXR columns, high efficiency for the treatment of a complex effluent containing a mixture of dyes and high salinity was observed. Data are unmatched with other studies because the conditions and aims of this research are different. In this study, an integrated membrane process for water recovery from an industrial effluent was used.

Numerous contributions show that adsorption is the most used process for the removal of dyes, exposing the different adsorbent capacity of innovative materials; however, the data are focused on the adsorption of a single component using synthetic solutions [41,42]. In addition, dynamic studies showing dye removal from industrial effluents are unusual. Here, rupture points are also different because they depend on adsorbent material, feed flow, dyes, dye concentration, and bed height or volume. Therefore, dissimilar adsorption data are included in these reports. Information on dynamic studies is found in Mustafa et al. [43]. The authors determined the maximum capacity of pinecone biomass as 239.9 mg/g for adsorption of 520 mg of methyl blue. Victor-Ortega et al. [44] used continuous flow to investigate the breakthrough time of an ion exchange column for the treatment of olive mill wastewater. The results were as follows: 147.5 min with 10 g/L of resin dosage to ensure 74% and 78% salt removal. Selambakkannu et al. [45] tested a bed column with prepared banana fibers for the removal of acid blue 80. The maximum bed capacity was 194.45 mg/g in adsorption conditions of 5 mL/min flow rate, 100 mg/L inlet concentration and 50 mm of bed height. Azoulay et al. [46] obtained data on the dynamic adsorption efficiency of a mixture of palm waste (100 mg) for methylene blue and methyl orange removal. The maximum capacities were 12.53 and 12.36 mg/g, respectively, using a flow rate of 1.5 mL/min.

In turn, IEXR are habitually suggested as a treatment prior to RO; principally when a high content of salts is detected in the effluents. In this case, the replacement of Na^+^ and Cl^−^ ions reduced salt concentration in the effluent, alleviating the RO operation. Therefore, the early fouling on the membrane was shortened, and superior RO efficiency was obtained in the water recovery process. In addition, IEXR were also integrated into the treatment process, because they are low cost and their regeneration is quick and easy, reducing the cost of the treatment.

Currently, IEXR are used in a combined process for a wide range of applications; however, the most common uses are water softening (removing calcium and magnesium ions), water demineralization (removal of all ions), and de-alkalization (removal of bicarbonates). Resin efficiency is measured by its capacity for ion removal; however, this parameter depends on the operating conditions, resin type, and the characteristics of the effluent (ions present), making it difficult to compare with other reports. Moreover, there have been few dynamic studies of cationic resins showing Na^+^ and Cl^−^ removal. Some interesting data were found in Millar et al. [47]. They treated an effluent with an initial Na concentration of 1259 mg/L, obtaining a maximum capacity for commercial cationic resin of 62.9 g Na/kg of resin. The rupture point was found after 29 L of processed water and 30 runs of effluent treatment at a flow rate of 6.9 L/h. Chandrasekara et al. [48] tested a mixture of commercial resins (cationic and anionic) to produce drinking quality water, removing salts from a saline effluent (0.1M). Using a continuous-flow system at a rate of approximately 10 mL/h and 9.6 mL of resins, they achieved six runs of treatment until rupture point. Li et al. [49] used ion exchange resins to remove Cl^−^ from industrial saline wastewater, containing 42,600–53,250 mg Cl/L. Equilibrium studies showed a resin capacity of 1800 mg Cl/g.

### 3.3. Membrane Efficiency in the Desalination Process of an Industrial Effluent

According to the treatment process design, the effluent from resin anionic II column was fed to the RO membrane to end the desalination and water recovery process. Therefore, information on membrane efficiency describes the effect of previous operations (S–AC–IEXR) on RO. Data on membrane operation correspond to flux profiles study and fouling dominance determination.

Figure 6 shows RO membrane behavior during the desalination process of industrial effluent treatment, exhibiting declines in water fluxes (permeation curves J, L/m^2^h). Membrane and salts removal efficiency by TDS removal (Rejection%) after 10 runs of the stability period of permeation during effluent treatment is also included; feed flow to RO: (a, b) 0.5 L/h; (c, d) 0.7 L/h; (e, f) 1 L/h.

Water flux decline represents the membrane permeation/water during the desalination of effluent for 300 min of operation time, using feed flows of 0.5, 0.7 and 1 L/h with a TDS range of 5120–7456 mg/L.

The profile of the flux curves exhibited typical behavior for flux permeate decline, displaying a gradual reduction in flux and an increment in TMP (>0.3 bar). This is due to membrane fouling, operation time, and the runs of effluent treatment causing the expected fouling.

In turn, Figure 6 depicts that flux permeate is affected by feed flow rate. The highest water flux reduction (60%) was observed when the membrane was fed at a flow of 1.0 L/min, indicating the highest membrane fouling caused by this flow. Consequently, a feed flow of 0.7 L/min was linked with low membrane fouling and high efficiency of salt removal (80–98%).

Water permeate fluxes exhibited a slight decrease of 10–20% for 100–150 min with permeation ranges of 28–20 L/h/m^2^. Afterward, a decrease of 30–50% was observed during the process of effluent treatment. The highest permeate flux (20–18 L/h/m^2^) was provided by a feed flow of 0.7 L/min, whereas feed flows of 1.0 L/min exhibited the lowest water permeability and reduced the amount of salts removal (30–40%); thus, an early fouling was observed for this feed flow rate.

Similar behavior was observed in rejection salts. Hence, it is observed that when membranes achieve 90% rejection, the permeate flux range reaches 15–25 L/h/m^2^, whereas when the rejection percentage is severally affected, permeate flux is <10 L/h/m^2^. In addition, feed flow also affected these data, showing a drastic reduction of salt rejection percentages.

According to flux profiles, a feed flow of 0.5–0.7 L/min was suggested as the most appropriate for effluent desalination, giving the highest membrane performance with high water fluxes and salt removal percentages.

Table 4 provides data describing the fouling predominance in the membrane during effluent desalination. Data include SDI and membrane resistances for feed flows of 0.5, 0.7, and 1.0 L/min, involving ranges of TDS in effluent feeds of 5120–7456 mg/L.

SDI < 1 in feed flows of 0.5 and 0.7 L/min indicate the influence of reversible fouling on water flux decline by salt deposition on the membrane surface. This is a result of salt content in the feed effluent.

Feed flows in the ranges of 0.5–0.7 L/min also confirmed reversible fouling dominance: R_r_ = 60%. This shows a maximum contribution in flux decline for 300 min of effluent desalination, whereas R_ir_ displayed irreversible fouling = 32% and hydraulic resistances 8%. Conversely, the effluent feed flow rate of 1 L/min caused SDI>1 and R_ir_ > R_r_, signifying the predominance of irreversible fouling.

The reversible fouling dominance described the cake layer formation on the membrane surface caused by salt rejection. The pores of the membrane are blocked by settled solids, organic dye and high salts concentration. The application of pretreatment reduced membrane blockage. In addition, membrane regeneration to initial hydrodynamic properties after each filtration operation cycle is also possible by hydraulic cleaning, which was indicated by R_m_ [50]. On the contrary, irreversible fouling predominance is normally associated with the blocking of membranes pores; thus, the membrane cannot return to its original condition, producing severe water flux decline and permanently low permeation, reducing membrane efficiency and membrane life.

The data demonstrate that alleviation of membrane fouling and fouling dominance are linked to the application of sequential operations prior to RO, consisting in S–AC–IEXR, providing a high efficiency in RO for water recovery, which involves high flux, fouling delay and reduction of this phenomenon. Particularly, results from fouling studies exhibited that AC was effective in eliminating colorants from the effluent, which reduced the presence of organic foulants from the RO membrane.

Adequate ranges of feed flows contributed to reversible fouling, indicating that pretreatment was an adequate strategy for RO alleviation because it mitigates fouling. Therefore, based on these effluent characteristics, the integrated membrane process S–AC–IEXT–RO is suggested as an alternative to the application of effluent treatment and water recovery.

Similar information on membrane efficiency in effluent treatment is available in Riera et al. [36]. Hernández et al. [5] studied an integrated process of UF–UF–RO as treatment of a food effluent, finding reversible fouling in RO with SDI < 1.9 and 15.4% of R_ir_, which was attributed to the presence of monovalent salts. Further, Güneş and Gönder [16] measured fouling dominance by membrane resistances as a result of the application of an integrated process of coagulation/flocculation and NF (EC–NF). Their data displayed that EC pretreatment achieved the reduction in the cake layer formation which demonstrated good agreement with resistance results.

Other publications showing fouling dominance in membrane processes are found in Antony et al. [51]. They compared fouling profiles of RO membranes in four municipal effluent samples and one industrial effluent sample. Scanning electron microscope (SEM) and energy-dispersive X-ray spectroscopy displayed that the dominant foulant in the industrial effluent was due to organic deposits. Khan et al. [52] carried out different tests and revealed that the fouling material on RO membranes is mainly composed of organic fouling, whereas, inorganic fouling is attributed to salt content, resulting in concentration polarization (CP).

Alternative pretreatments to alleviate fouling on RO membranes can be seen in Yin et al. [53]. The authors treated a textile effluent using operations prior to RO (coagulation–ion exchange and UF–ion exchange) (C–IEXR–UF–IEXR) to reduce membrane fouling. Due to organic component removal, RO was highly efficient and is a prominent method of water recovery. Shinde et al. [54] used an integrated process (adsorption, ion exchange resins, and RO membrane) (A–IEXR–RO) for the treatment of saline wastewater collected from different sewage sources, showing a maximum flux of 34 L/h/m^2^ and no early indication of membrane fouling. Chen et al. [15] compared the application of adsorption (A) and coagulation (C) as pretreatment methods for wastewater depuration, determining their effect on RO membrane fouling. The most effective process was coagulation; however, the fouling type was intermediate blocking. Similarly, Zhang et al. [12] tested UF and integrated electrochemical–microfiltration–granular activated carbon adsorption (e–MF–GAC) as pretreatment processes to control RO fouling. The most effective process for removing inorganic/organic foulants and maintaining high water fluxes in RO membranes was e–MF–GAC.

Specifically, data on food wastewater treatment are scarce; a study on rinse water treatment for water recovery was found in [40]. Authors used an integrated process based in NF-RO membrane operations. In this case, flux decline after 60 min of membrane operation was observed and reversible fouling predominance was detected during effluent treatment, indicating a effective strategy for water and milk solid recovery. However, results may vary as a consequence of operating conditions and membrane characteristics.

In turn, RO has been addressed in several preceding reports on salt removal as the only operation of brine effluents. Currently, this technology is also the most widely used method for desalination, and it is also the most used at the industrial level. However, fouling-resistant membranes remain a challenge, as described in recent reviews [55,56].

## 4. Conclusions

Treatment of industrial effluents for water recovery is an alternative to water supply in the industrial sector, which is experiencing water scarcity and emits polluted effluents. There exists the possibility of conversion of contaminated effluents to potable water.

The present report discussed the design and evaluation of an integrated membrane process for potable water recovery by treatment of an industrial effluent stemming from the manufacture of food additives.

Because the principal contents of the effluent were red colorant mixture and inorganic salts, the integrated membrane process consisted of four operations prior to RO, involving sedimentation for settleable and suspended solids removal and adsorption by AC for the removal of organic compounds (dyes mixture). IEXR were used for salt reduction, using cationic and anionic resins (one and two columns respectively), eliminating Na^+^ and Cl^−^ from the effluent to reduce fouling on the RO membrane, and increasing the efficiency of clean water production.

The sequential and integrated treatment as well as the operating conditions allowed the production of potable water, increasing the efficiency of membrane process. Clean water satisfied the permissible limit for TSD, COD, and EC for potable water. The dynamic studies of adsorption and ion exchange columns showed the effectiveness of these operations in an integrated RO process, exhibiting high flux and reversible fouling as principal dominance.

Particularly, packed AC columns showed high efficiency for color elimination, reaching rupture point at 20 h and after of 5L of effluent treatment. IEXR showed capacity for salt removal, providing 2–2.5 L of effluent treatment at 11–15 h of rupture point.

The RO membrane exhibited a flux range of 28–20 L/h/m^2^ for 100–150 min. A reduction in flux (30–40%) was observed later, maintaining reversible fouling dominance at a feed flow of 0.5–0.7 L/h.

The results indicated that the integrated membrane process was suitable for effluent treatment and water recovery, providing an excellent performance of the RO membrane by mitigating fouling. Sedimentation enhanced adsorption by AC and IEXR operation. Consecutively, adsorption by AC was appropriate for the removal of organic colorants and fouling reduction in IEXR and RO membrane. In turn, IEXR removed ions from inorganic salt, such as Na^+^ and Cl^−^, reducing fouling in RO caused by the presence of Cl^−^ and stable salts.

According to dynamic studies of adsorption and IEXR, large rupture points were obtained, showing technic viability of implementation on a large scale. In turn, the integrated process S–AC–IEXR–RO is energetically feasible, because previous operations to RO do not require power to run. Therefore, the described integrated process could be scaled up, and its effectiveness for water recovery in the treatment of industrial effluents with similar characteristics could be adapted and enhanced.

## Figures and Tables

**Figure 1 membranes-12-00742-f001:**
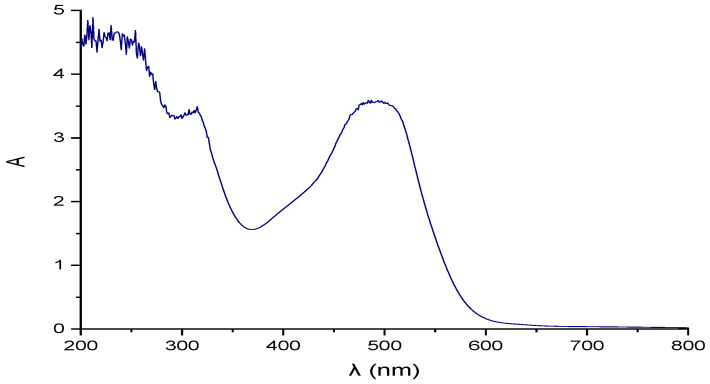
Industrial effluent coloration and Uv-vis spectra (200–800 nm).

**Figure 2 membranes-12-00742-f002:**
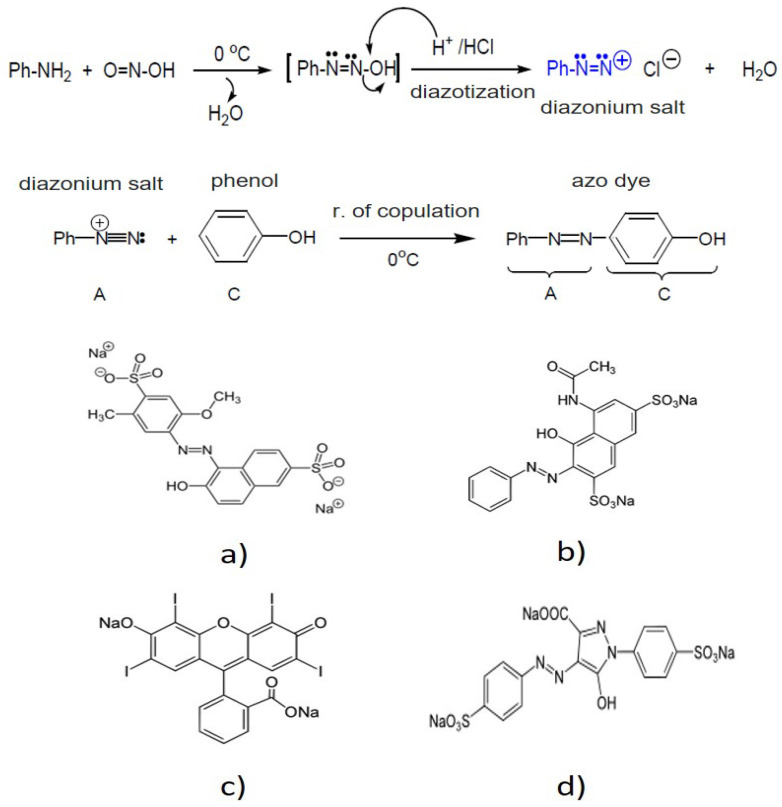
Chemical reactions involved in the production of azo dyes, and chemical structure of dyes. (**a**) Red 40, (**b**) Red 2, (**c**) Red 3, (**d**) Yellow 5.

**Figure 3 membranes-12-00742-f003:**
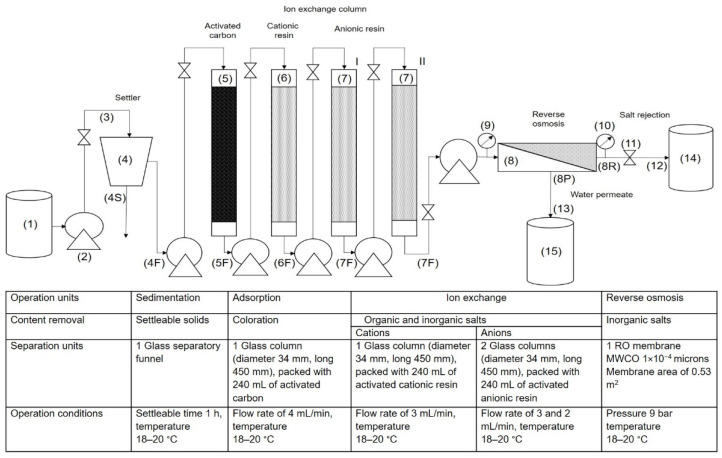
Diagram of the treatment process of industrial effluent treatment for water recovery.

**Figure 4 membranes-12-00742-f004:**
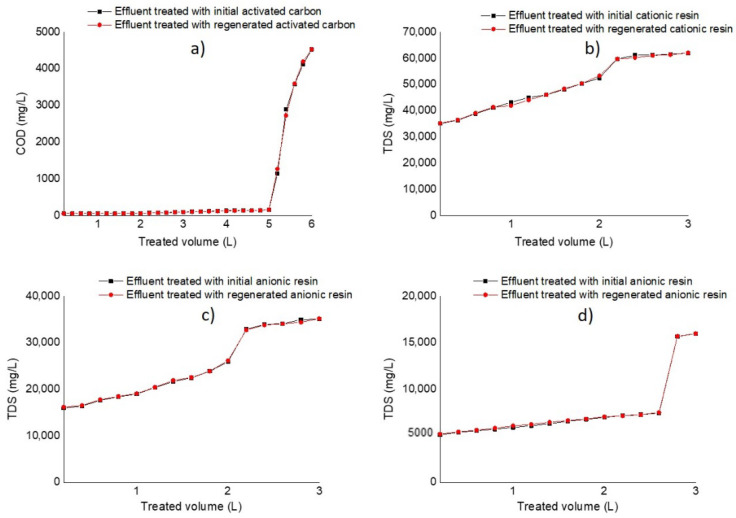
Rupture curves of materials packaging in adsorption and ion exchange columns for 30 and 15 runs of effluent treatment at initial conditions and after their regeneration. (**a**) Activated carbon, (**b**) cationic resin, (**c**) anionic resins in column I, (**d**) anionic resin in column II.

**Figure 5 membranes-12-00742-f005:**
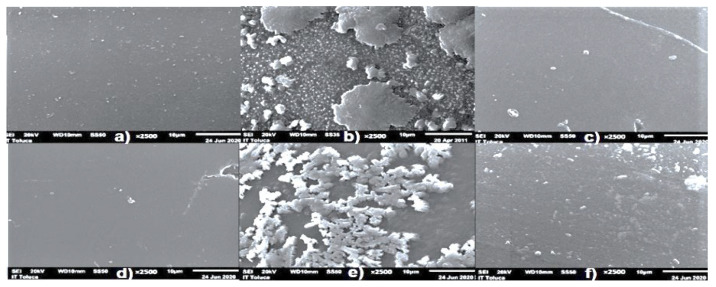
Micrographs at ×2500 of the sample surface resins. Cationic: (**a**) initial, (**b**) saturated, (**c**) regenerated. Anionic: (**d**) initial, (**e**) saturated, (**f**) regenerated.

**Figure 6 membranes-12-00742-f006:**
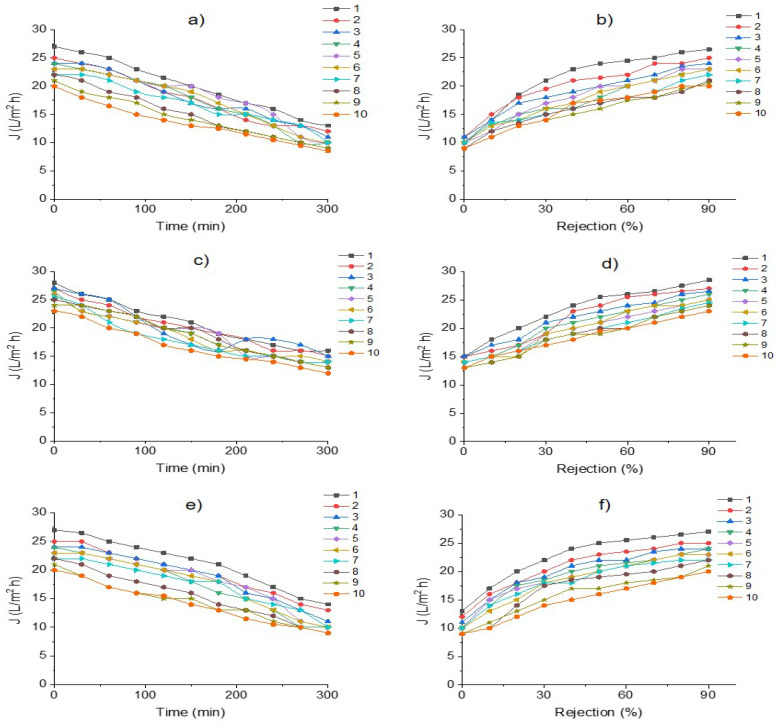
**Figure 6** shows RO membrane behavior during the desalination process of industrial effluent treatment, exhibiting declines in water fluxes (permeation curves J, L/m^2^h). Membrane and salts removal efficiency by TDS removal (Rejection%) after 10 runs of the stability period of permeation during effluent treatment is also included; feed flow to RO: (**a**,**b**) 0.5 L/h; (**c**,**d**) 0.7 L/h; (**e**,**f**) 1 L/h.

**Table 1 membranes-12-00742-t001:** Physicochemical characteristics of the industrial effluent.

Parameter	pH	COD (mg/L)	EC (mS/cm)	Cl^−^ (mg/L)	SS (mL/L)	TDS (mg/L)	TS (mg/L)	Red Coloring Absorbance at 490 nm
Parameter value	9 ± 0.1	4640 ± 100	87.5 ± 15	36,301 ± 125	190 ± 0.5	66,100 ± 150	66,300 ± 550	3.5

**Table 2 membranes-12-00742-t002:** Physicochemical characteristics of the effluent treated through the different unit operations of the treatment process of industrial effluent.

Parameters	Industrial Effluent	Treated Industrial Effluent						Requirements of Industry and Potable Water Standard Regulations
		Operation 1 Sedimentation (S)	Operation 2 Adsorption by AC	Operation 3 IEXR			Operation 4 RO Membrane(Permeate)	
				Cationic	Anionic I	Anionic II		
pH	9 ± 0.1	8.06 ± 0.05	8.5 ± 0.1	7.8 ± 0.1	8.2 ± 0.2	8.5 ± 0.1	7.6 ± 0.2	6–9
EC (mS/cm)	87.5 ± 15	86.5 ± 0.01	82.1 ± 0.02	43.8 ± 0.01	35.9 ± 0.01	18 ± 0.2	0.64 ± 0.1	<1.8
Cl^−^ (mg/L)	36,301 ± 125	36,301 ± 140	30,558 ± 140	18,184 ± 70	8345.5 ± 20	3435 ± 15	207 ± 70	<250
COD (mg/L)	4640 ± 100	4530 ± 100	56 ± 20	98 ± 20	71 ± 2	24.6 ± 2	N. D	Undefined
TS (mg/L)	66,300 ± 550	66,260 ± 200	63,162 ± 200	35,685 ± 100	16,240 ± 150	5100 ± 15	128 ± 2	<1000
TDS (mg/L)	66,100 ± 150	66,000 ± 100	62,050 ± 200	35,105 ± 100	15,985 ± 150	5120 ± 15	125 ± 2	<300
SS (mL/L)	190 ± 0.5	*ND	ND	ND	ND	ND	ND	<10
Red coloring Absorbance	3.5 10% dilution	3.5 10% dilution	ND	ND	ND	ND	ND	Uncolored

*ND = Non detect value.

**Table 3 membranes-12-00742-t003:** Data of removal contaminants capacity of packaging materials (AC and IEXR) during industrial effluent treatment in columns with 20 cm of bed height.

Packaging Materials	AC (COD)	Cationic IEXR (TDS)	Anionic IEXR-I (TDS)	Anionic IEXR-II (TDS)
Runs of effluent treatment	30	15	15	15
C_o_ (mg/L)	4530 ± 100	62,050 ± 200	35,105 ± 100	15,985 ± 150
C_b_ (mg/L)	150 ± 50	52,500 ± 100	25,907 ± 150	7456 ± 120
V_b_ (L)	5 ± 0.2	2 ± 0.5	2 ± 0.5	2.5 ± 0.5
θ_b_ (h)	20 ± 3	11 ± 2	11 ± 2	15 ± 2
Q (mL/min)	4 ± 0.3	3 ± 0.3	3 ± 02	2 ± 0.2
q_b1_ (mg/mL)	91 ± 1	80 ± 1	77 ± 1	88 ± 1
q_b2_ (mg/g)	3425 ± 100	8531 ± 100	5408 ± 100	2985 ± 100

**Table 4 membranes-12-00742-t004:** Data of SDI and resistances of flux permeation (nm^−1^) from RO membrane during effluent treatment.

Parameters	Feed Flow (L/min)		
	0.5	0.7	1.0
SDI	0.98 ± 0.1	0.97 ± 0.1	1.1 ± 0.1
R_t_	43 ± 1	45 ± 1	55 ± 1
R_m_	5 ± 0.1	5 ± 0.1	5 ± 0.1
R_ir_	14 ± 1	14 ± 1	35 ± 1
R_r_	24 ± 1	25 ± 1	15 ± 1

## Data Availability

Not applicable.

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
