# Peer review of "Membrane Water Treatment for Drinking Water Production from an Industrial Effluent Used in the Manufacturing of Food Additives"

_membranes, 2022, doi:10.3390/membranes12080742_

Round 1

Reviewer 1 Report

This paper summarized potable water production by a membrane treatment process of an industrial effluent from the manufacture of food additives. The authors must address the newness of this study in comparison of other studies. It is felt that the application of the processes mentioned are not new, hence lacking originality. It is possible for the manuscript to be accepted after major corrections.  

Main comments:

Comment 1: The article topic is a bit misleading in a sense that it was not a membrane treatment process alone. It was an integrated system where membrane is one of the treatments used to treat the industrial effluent. The authors should amend the article topic to fully capture the said study. The introduction can be further improved with more references on the processes involved. Justifications are essential for the chosen processed to treat the wastewater. 

Comment 2: A graphical abstract is suggested.

Comment 3: There are a lot of inconsistencies, a lot of grammatical and technical mistakes hence could potentially lead to misrepresentation. Few examples as follows:

       i.          Line 66: spelling mistake “matter”.

      ii.          Line 69: “available” repeated twice in sentence.

    iii.          Line 84-85: The authors need to explain for the materials used. Acronyms need to be spelled out, formulae need to be edited accordingly. This applies to all in the manuscript.

    iv.          Line 94-98: The instruments must be properly mentioned and the parameters that it is measuring. The use of potentiometer was mentioned but there was no explanation for its use.

      v.          Line 118, 123, 129, 130, Figure 2: Food azo “days”? This spelling mistake is seen in other parts of the manuscript. Please check the spelling in all sections.

Comment 4: Quality of the Figures (1 and 2) can be further improved. The details of the image are not clear and difficult to read.

Comment 5: Line 112. How did the authors characterize the colorant mixtures? In this case red No 2, red no 3, red no 40 and yellow no 5. It was not clearly mentioned how these mixtures were characterized?

Comment 6: Section 2.6. Statistical analysis was mentioned in this section. However the purpose was not elaborated further. Not entirely sure if Section 2.6 is appropriate.

Comment 7: Figure 4 – The x axes were not labelled properly. The authors could improve the figure better to visualize between the 2 parameters, i.e. volume of treated effluent and operation time. It was also not mentioned clearly how the resins were regenerated. How were the ions separated from the resins?

Comment 8: Figure 6 – The figure was not properly captioned. It was labelled (a) until (f) but there was no information or description about them in the caption and also in the text. The authors must clearly mentioned them in the caption of Figure 6 and also in the text.

Comment 9: Table 4 – How was SDI calculated. It was not explained clearly on its significance. The argument is weak. The authors must provide references to support their justifications.

Comment 10: The conclusion section lacked substantial justifications on the gathered results. This must be addressed by the authors to fully capture what was achieved from the conducted experiments.  Overall, there must be major revamping of the manuscript for it to be considered for publication.

Author Response

Manuscript ID: membranes-1775271

Last title: Potable water production by a membrane treatment process of an

industrial effluent from the manufacture of food additives

New title:  MEMBRANE WATER TREATMENT FOR DRINKING WATER PRODUCTION FROM AN INDUSTRIAL EFFLUENT USED IN THE MANUFACTURING OF FOOD ADDITIVES.

Reviewe 1

Comment 1: The article topic is a bit misleading in a sense that it was not a membrane treatment process alone. It was an integrated system where membrane is one of the treatments used to treat the industrial effluent. The authors should amend the article topic to fully capture the said study. The introduction can be further improved with more references on the processes involved. Justifications are essential for the chosen processed to treat the wastewater.

Response: We have included more authors and have made several modifications to the writing so that it is understood that the article contains the evaluation of a system composed of several operations including a RO membrane and that the function of the operations prior to RO is the mitigation of fouling in the membrane.

Comment 2: A graphical abstract is suggested.

Response: Authors include a graphical abstract

Comment 3: There are a lot of inconsistencies, a lot of grammatical and technical mistakes hence could potentially lead to misrepresentation. Few examples as follows:

  1. Line 66: spelling mistake “matter”.

Response: the matter word has been eliminated of the manuscript.  

  1.           Line 69: “available” repeated twice in sentence.

Response: Line 69 was changed.

    iii.          Line 84-85: The authors need to explain for the materials used. Acronyms need to be spelled out; formulae need to be edited accordingly. This applies to all in the manuscript.

Response: The explication of used materials was included; Acronyms are also included in the manuscript.

  1. Line 94-98: The instruments must be properly mentioned and the parameters that it is measuring. The use of potentiometer was mentioned but there was no explanation for its use.

Response: Line 187-193 were included to complement the information

  1. Line 118, 123, 129, 130, Figure 2: Food azo “days”? This spelling mistake is seen in other parts of the manuscript. Please check the spelling in all sections.

Response: All the manuscript was reviewed and corrected

Comment 4: Quality of the Figures (1 and 2) can be further improved. The details of the image are not clear and difficult to read.

Response: Figure 1 and were changed.

Comment 5: Line 112. How did the authors characterize the colorant mixtures? In this case red No 2, red no 3, red no 40 and yellow no 5. It was not clearly mentioned how these mixtures were characterized?

Response: According the effluent origin and Uv-Vis spectra of effluent, was suggested the presence of red No 2, red no 3, red no 40 and yellow no 5. However, Uv- vis spectra of the mixture of these colorants confirmed the suggested presence.   Line 213-220 explain as the mixture was characterized.

Comment 6: Section 2.6. Statistical analysis was mentioned in this section. However, the purpose was not elaborated further. Not entirely sure if Section 2.6 is appropriate.

Response: Title of section 2.6 was removed

Comment 7: Figure 4 – The x axes were not labelled properly. The authors could improve the figure better to visualize between the 2 parameters, i.e. volume of treated effluent and operation time. It was also not mentioned clearly how the resins were regenerated. How were the ions separated from the resins?

Response: Figure 4 of the new version of manuscript only contain a only x axe.  

Comment 8: Figure 6 – The figure was not properly captioned. It was labelled (a) until (f) but there was no information or description about them in the caption and also in the text. The authors must clearly mentioned them in the caption of Figure 6 and also in the text.

Response: lines 625-629 indicate the corrected caption of Figure 6. The correction of caption of Figure 6, was also placed at the bottom of the figure.   

Comment 9: Table 4 – How was SDI calculated. It was not explained clearly on its significance. The argument is weak. The authors must provide references to support their justifications.

Response: Reference of SDI determination is indicated in line 362.  

Comment 10: The conclusion section lacked substantial justifications on the gathered results. This must be addressed by the authors to fully capture what was achieved from the conducted experiments.  Overall, there must be major revamping of the manuscript for it to be considered for publication.

Response: Line 747-757 were included in conclusion section to provide a substantial justification of the investigation.  

Reviewer 2 Report

Membranes- MDPI

Potable water production by a membrane treatment process of an industrial effluent from the manufacture of food additives

A bench scale of water treatment process was used to treat wastewater industrial food additives effluent

The manuscript is discussing an important issue. However, there are some comments and suggestions for the authors to consider before publication.

·       Short abstract with insufficient findings.

·       The introduction is short. Please, provide more information and include all relevant references.

·       The authors should clarify the problem and the novelty of their study.

·       Please, rewrite section 2.1

·       In section 2.2, some of effluent characteristics were analyzed, what about:

Temperature, pH, BOD, total N, dissolved oxygen

·       Kindly, illustrate the type of RO (low pressure or high pressure) and the amount of pressure you used in the experimental part.

·       Check the unit format.

·       Check line 207 (what do you mean by calculi)?

·       Add some information about reactivation of the adsorption (AC) and ion exchange resins.

·       Treatment cost is important factor. I think the water treatment cost of your study will be higher than the conventional methods. Please calculate it.

·       Long term cost of ion exchange is high, I can’t see a good reason to use it:

o   Try to exclude it and measure the treated water characteristics.

o   Use alternative technologies ( e.g. NF, to reduce the fouling on the RO membrane)  instead of ion exchange. Then, measure the treated water characteristics.

·       Add some results in conclusion.

Author Response

Manuscript ID: membranes-1775271

Last title: Potable water production by a membrane treatment process of an

industrial effluent from the manufacture of food additives

New title:  MEMBRANE WATER TREATMENT FOR DRINKING WATER PRODUCTION FROM AN INDUSTRIAL EFFLUENT USED IN THE MANUFACTURING OF FOOD ADDITIVES.

Reviewer 2

  1. Short abstract with insufficient findings.

Response: According to suggestion, abstract was modified

  1. The introduction is short. Please, provide more information and include all relevant references.

Response: The introduction was enlarged, including other authors. Line 137-162 contains the changes.     

  1. The authors should clarify the problem and the novelty of their study.

Response: Line 168-173 clarify the problem of the study

4· Please, rewrite section 2.1

Response: Section 2.1 include the use of reagents

  1. In section 2.2, some of effluent characteristics were analyzed, what about: Temperature, pH, BOD, total N, dissolved oxygen

Response: Industrial effluent was characterized according its origin and requirements for reuse. pH was included in parameters; however other parameters as BOD and total N, were no included, because of origin of effluent is the manufacture of synthetic dyes.     

  1. Kindly, illustrate the type of RO (low pressure or high pressure) and the amount of pressure you used in the experimental part.

Response: Line 269 indicates the Ro pressure.

  1. Check the unit format.

Response: The manuscript was review and the format was checked

  1. Check line 207 (what do you mean by calculi)?

Response: Line 219-221 contains information on bed volume and bed mass.

  1. Add some information about reactivation of the adsorption (AC) and ion exchange resins.

Response:  Line 328-333 includes information on activated carbon and resins regeneration

10· Treatment cost is important factor. I think the water treatment cost of your study will be higher than the conventional methods. Please calculate it.

Response: Authors decide not to include the cost of suggested treatment in the present manuscript, because, economic, environmental and technical viability of integrated process of treatment are part of other manuscript.  

  1. Long term cost of ion exchange is high, I can’t see a good reason to use it:

Response: All treatment processes have advantages and disadvantages; including membranes technology, because they also present high cost. However, when the process is designed to solve real problems of industry of water recovery, the cost is minimum in comparison with the environmental gain.

Also, when a process is suggested as treatment should make sure its implementation, maintenance and raw material. Therefore, the process includes conventional methods, because they are known.  

In the case, the main advantage of ion exchange resins its easy and rapid regeneration. This is a reason for wide use in industry. Commercial system of water treatment, also include these processes accompany RO to mitigate the fouling.

  1. Try to exclude it and measure the treated water characteristics.

Response: We did not understand this observation

  1. Use alternative technologies (e.g. NF, to reduce the fouling on the RO membrane) instead of ion exchange. Then, measure the treated water characteristics.

Response: NF also reduce the fouling of the membrane; however, this membrane also present fouling, and the cost the process is incremented.   

  1. Add some results in conclusion.

Response: some results were included in line 758-768.  

Round 2

Reviewer 1 Report

Thank you for the updated version. The manuscript has improved a lot better than the previous version. The authors have addressed the issues raised.

Minor comments:

Comment 1: In the Introduction section, the authors should explain the missing knowledge gap in implementing the chosen system. What is novel and new in this chosen system to treat the industrial effluent in comparison with other systems reported elsewhere?

 Comment 2: In the Conclusion section, the authors should explain briefly how the process could be scaled up – explain the effect in terms of energy consumption and efficiency of the system.

Author Response

Manuscript ID: membranes-1775271

Last title: Potable water production by a membrane treatment process of an industrial effluent from the manufacture of food additives

New title:  MEMBRANE WATER TREATMENT FOR DRINKING WATER PRODUCTION FROM AN INDUSTRIAL EFFLUENT USED IN THE MANUFACTURING OF FOOD ADDITIVES.

Reviewer 1

  1. In the Introduction section, the authors should explain the missing knowledge gap in implementing the chosen system. What is novel and new in this chosen system to treat the industrial effluent in comparison with other systems reported elsewhere?

Response: Authors inserted in introduction and section 3.1 the purpose system used as treatment of industrial effluent to water recovery.   

  1. In the Conclusion section, the authors should explain briefly how the process could be scaled up – explain the effect in terms of energy consumption and efficiency of the system. Short abstract with insufficient findings.

Response: Authors inserted some lines to address the observations  

Reviewer 2 Report

The authors have addressed all comments and I am satisfied with the improvements.

Author Response

There are not comments